# Enzymatic Control of Alcohol Metabolism in the Body—The Roles of Class I, II, III, and IV Alcohol Dehydrogenases/NADH Reoxidation System, Microsomal Ethanol Oxidizing System, Catalase/H_2_O_2_ System, and Aldehyde Dehydrogenase 2

**DOI:** 10.3390/ijms26199479

**Published:** 2025-09-27

**Authors:** Takeshi Haseba

**Affiliations:** Department of Legal Medicine, Graduate School of Dentistry, Kanagawa Dental University, 82 Inaokacho, Yokosuka 238-8580, Japan; haseba@kdu.ac.jp or hasebat@nms.ac.jp

**Keywords:** alcohol metabolism, Class I ADH1, Class II ADH2, Class III ADH3, Class IV ADH4, NADH reoxidation rate, catalase, MEOS, ALDH2

## Abstract

Alcohol metabolism in the body is a key theme in medical research on alcohol. It is primarily regulated by the alcohol dehydrogenase (ADH) and mitochondrial NADH reoxidation in the liver. Class I ADH1 is a well-known ADH isozyme and a key enzyme in alcohol metabolism, with the lowest Kms for ethanol and the highest sensitivity to pyrazole (Pz) among the ADH isozymes. However, a Pz-insensitive metabolic pathway also plays a role in systemic alcohol metabolism, with increasing metabolic contributions at higher blood alcohol concentrations (BACs) and under chronic alcohol consumption (CAC). The Pz-insensitive pathway is referred to as the non-ADH pathway—specifically, it is a non-ADH1 pathway—and is assumed to involve the microsomal ethanol oxidizing system (MEOS) or catalase, as both enzymes are insensitive to Pz and exhibit higher Kms than ADH1. The MEOS is a favored candidate for this pathway, as its activity markedly increases with the rate of alcohol metabolism under CAC. However, the role of the MEOS in alcohol metabolism has not been proven in vivo (even under CAC conditions), nor has that of catalase. Here, we report Class III ADH3 as a new candidate in the non-ADH1 pathway, as it also has a lower sensitivity to Pz and a higher Km. It is markedly activated by lowering Km following the addition of amphiphilic substances, which increases the solution’s hydrophobicity in the reaction medium; additionally, Nile red staining demonstrates a higher solution hydrophobicity in the cytoplasm of mouse liver cells. The rate of alcohol metabolism in *ADH1* knockout (*Adh1*^−/−^) mice—which depends solely on the non-ADH1 pathway—increased by more than twice under CAC and was significantly correlated with the amount of liver ADH3 protein, but not with CYP2E1 protein (a main component of the MEOS). The rate of alcohol metabolism in *Adh3*^−/−^ mice lacking ADH3 decreased in a dose-dependent manner compared with wild mice. The liver ADH3 protein in wild-type mice increased in line with the ADH1 protein under CAC. These data suggest that ADH3 contributes to alcohol metabolism in vivo as a non-ADH1 pathway and to the enhancement of alcohol metabolism under CAC through activation of the ADH1/ADH3/NADH reoxidation system. In alcoholic liver diseases, ADH1 activity decreased with the progression of liver disease, while ADH3 activity increased or was maintained even in alcoholic liver cirrhosis. Therefore, the role of ADH3 in alcohol metabolism may be increased in the context of alcoholic liver diseases, complementing the reduced role of ADH1. It has also been suggested that Class II ADH2, Class IV ADH4, and aldehyde dehydrogenase (ALDH) 2 play roles in alcohol metabolism in vivo under certain limited conditions. However, ADH2 and 4 may not contribute to the enhancement of alcohol metabolism through CAC.

## 1. Introduction

Studies on alcohol metabolism, which elucidate the enzymatic mechanisms underlying alcohol metabolism in the body, are fundamental in medical research on alcohol. Ingested ethanol (EtOH) is oxidized to acetaldehyde (AcH) by alcohol dehydrogenase (ADH), and AcH is then oxidized to acetic acid by aldehyde dehydrogenase (ALDH), mainly in the liver. In both enzyme reactions, the coenzyme NAD is reduced to NADH, which is then reoxidized to NAD in the mitochondrial respiratory chain and used again to oxidize EtOH and AcH. This series of processes is known as alcohol metabolism. Acetic acid is then incorporated into physiological metabolism, such as the TCA cycle and fatty acid synthesis, in the form of acetyl-CoA.

Research on alcohol metabolism has a long history [1,2] and has been conducted mainly by focusing on ADH in the liver [3,4]. The enzyme initially called ADH is specifically inhibited by pyrazole (Pz), has low Kms for EtOH (~1~ mm), reaches Vmax at around 15 mm [5], and is currently classified as Class I ADH1 [6,7]. However, the rate of alcohol clearance from human blood (β value) is known to increase even above the blood alcohol concentration (BAC) at which the ADH reaches Vmax [8]. Furthermore, there is an alcohol metabolic pathway that remains even after the administration of Pz, the role of which increases under high BAC and/or chronic alcohol consumption (CAC) [9,10]. This type of alcohol metabolism cannot be explained by the enzymatic properties of this ADH; therefore, it is commonly called the non-ADH pathway [10], which specifically refers to the non-ADH1 pathway. The nature of this pathway has been the subject of prominent and heated debate throughout the history of alcohol medical research, and its definition has been expanded to include the microsomal EtOH oxidation system (MEOS) [9,10,11] and the peroxisome H_2_O_2_-dependent catalase EtOH oxidation system [12,13], as neither is inhibited by Pz and both have higher Kms than ADH [14,15]. Due to the large number of studies that have been conducted on these systems, reviews and textbooks currently state that "alcohol metabolism is carried out not only by ADH but also by MEOS and catalase" (Figure 1). As a result, the discourse suggesting that the enzymatic mechanism underlying alcohol metabolism has already been elucidated became dominant, and research on alcohol metabolism has decreased. However, the true nature of the non-ADH (i.e., ADH1) pathway has not been determined, as no clear evidence has been obtained in vivo for either catalase or the MEOS. During the previous debate on these two candidates, ADH isozymes other than ADH1 did not receive much attention, even though Class II ADH2 [16], Class III ADH3 [17,18], and Class IV ADH4 [19] were identified later and shown to satisfy criteria for being part of non-ADH1 pathways;, i.e., higher Kms and lower sensitivity against Pz inhibition compared with ADH1 (Table 1). Among them, we discovered ADH3 in mouse liver and found it to be the least inhibited by Pz and to have the highest Km among all the ADH isozymes [20]; accordingly, we have studied it as a candidate for the non-ADH1 pathway [21,22,23,24].

The lineup of alcohol-metabolizing enzymes is now nearly complete: Class I, II, III, IV, V [25], VI [26] ADHs, the MEOS, catalase, and ALDH2. In this paper, we review the literature—including our previous research—to understand under what conditions (and how) these enzymes contribute to in vivo alcohol metabolism; additionally, we provide an overview of how the enzymatic mechanisms underlying alcohol metabolism have been elucidated.

**Table 1 ijms-26-09479-t001:** Human alcohol dehydrogenase (ADH) isozymes.

Class	Isozyme Name	Synonym Name	Subunit	Gene Name		Km (EtOH)	Pyrazole Sensitiviy	Distribution	Original Publication
				Duester et al. [27]	Edenberg	(mM)			
I	ADH1A	ADH1	α	*ADH1A*	*ADH1A*	4.0	+++++	Liver	Wartburg et al., 1964 [5]
	ADH1B	ADH2	β1, β2, β3	*ADH1B*	*ADH1B*	0.05–40		Kidney, Stomach,	
	ADH1C	ADH3	γ1, γ2	*ADH1C*	*ADH1C*	0.6–1.0		Small intestine, etc.	
II	ADH2	ADH4	π	*ADH2*	*ADH4*	30	++	Liver	Li et al., 1977 [16]
III	ADH3	ADH5	χ	*ADH3*	*ADH5*	>1000	+	Liver, Ubiquitous	Haseba et al.,1979 [17] (mouse)
									Pares et al., 1981 [18]
IV	ADH4	ADH7	σ	*ADH4*	*ADH7*	30–580	++	Stomach, Esophagus	Pares et al., 1990 [19]
V	ADH5	ADH6	undetected	*ADH5*	*ADH6*	?	?	Liver, Stimach	Yasunami et al.,1991 [25]
VI	ADH6		undetected	*ADH6*		?	?	Unknown	Hoog et all., 2001 [26] (rat)

Modified from Duester et al. (1997) [27], and Edenberg (2007) [28], and Haseba [23]. mM; milli mol of EtOH, + ~ +++++; Inhibition sensitivity of ADH isozyme by Pyrazole.

## 2. The Fundamental System of Alcohol Metabolism via Liver ADH/NADH Reoxidation

### 2.1. ADH Activity in Liver Extract

When liver ADH activity is measured using mammalian liver extract as an enzyme sample, the activity reaches Vmax at approximately 15 mM of EtOH and is inhibited by more than 90% by Pz. Purified ADH also exhibits Vmax at approximately 15 mM of EtOH and is completely inhibited by Pz; therefore, ADH activity in liver extract is considered to be caused by a single enzyme (e.g., ADH1). It has been known since 1977 that, in addition to ADH1, liver extracts also contain ADH2 [16] and ADH3 [17,18]. However, measuring ADH activity in liver extract via the conventional ADH activity assay using low concentrations of EtOH as a substrate is thought to mainly reflect ADH1 activity, because these new ADH isozymes have higher Kms for EtOH compared with ADH1 and are less inhibited by Pz. Multiple reports have suggested that the ADH activity level in liver extract is the rate-limiting factor for in vivo alcohol metabolism based on measurements of the β value (mg/mL/h) or body alcohol elimination rate (AER, mg/kg/h) [29,30,31,32,33]. On the other hand, multiple reports also suggest that ADH activity in liver extract is not the rate-limiting factor for in vivo alcohol metabolism [34,35,36,37,38]. Indeed, the rate of alcohol metabolism shows a strong positive correlation with ADH activity in liver extract up to a certain level [29,33], above which the rate reaches a ceiling even if activity increases, showing a poor correlation with enzyme activity [39]. These reports indicate that other rate-limiting factors besides liver ADH activity—e.g., the NADH reoxidation rate—are involved in in vivo alcohol metabolism when liver ADH activity is sufficiently high, as described later.

There are many reports on the fluctuation in ADH activity in liver extract under CAC. DeCarli and Lieber [40] set up a CAC experiment in which rats were fed a liquid diet containing 5% EtOH (13 g/kg/day) for approximately one month. This CAC experiment resulted in alcoholic fatty liver in rats independently of nutritional factors and elevated alcohol metabolism and hepatic MEOS activity [11]. This Liber–DeCarli liquid diet (LD) method has become the standard for CAC animal experiments. As ADH activity in liver extract has been shown to either decrease or remain unchanged in LD experiments [9,41,42], it is generally believed that ADH does not contribute to the elevated alcohol metabolism caused by CAC [43]. On the other hand, several studies have demonstrated increased ADH activity in liver extract under CAC without utilizing the LD method [44,45]. We previously reported that the ADH activity in liver extracts of mice increased even under CAC with the LD method when the EtOH concentration in the LD was reduced to 4% [22,46]. Additionally, in rats, ADH activity in liver extracts increased under CAC under conditions of mild or no liver damage, but decreased as liver damage progressed [47,48]. In humans who are alcoholics, ADH activity in liver extract tends to be high in those with mild liver damage but decreases in those with severe liver damage, such as cirrhosis [49,50]. Thus, ADH activity in the liver is thought to initially increase under CAC but decrease as liver damage progresses.

### 2.2. NADH Reoxidation Rate

During alcohol metabolism, the coenzyme NAD is reduced to NADH through the EtOH oxidation reaction catalyzed by ADH, which is localized in the cytosol of hepatocytes. For ADH to continuously oxidize EtOH in hepatocytes, it is necessary to transport NADH from the cytosol to mitochondria, oxidize it via the respiratory chain, and supply the reoxidized NAD to ADH. Therefore, in ADH-dependent alcohol metabolism, both the hepatic ADH activity level and NADH reoxidation rate are considered rate-limiting factors. The NADH reoxidation rate is controlled via two steps: the activity of the NADH shuttles (e.g., malate–aspartate shuttle, α-glycerophosphate cycle) that transport cytosolic NADH to mitochondria, and the activity of the respiratory chain that oxidizes NADH in mitochondria. For example, the rate of alcohol metabolism decreases due to a deficiency of malate, aspartate, and α-glycerophosphate, which are substrates necessary for shuttle activity [36], or due to a decrease in the activity of aspartate aminotransferase or α-glycerophosphate dehydrogenase, which use these shuttle substrates to transport NADH into mitochondria [51,52]. However, the conditions under which the NADH shuttle becomes a rate-limiting factor during alcohol metabolism may be limited to in vitro studies: a leakage of shuttle substrates is observed when using isolated hepatocytes, but this may not occur in vivo [29,53]. In another step in NADH reoxidation, the activity of the mitochondrial respiratory chain also regulates the rate of alcohol metabolism; the rate decreases as the activity decreases [51,54,55,56,57] and increases as the activity increases [58,59,60].

Comparing the rate of alcohol metabolism and hepatic ADH activity between animal species revealed that the rate of alcohol metabolism in mice (~550 mg/kg/h) is more than five times that in humans (100 mg/kg/h) [37], but there is no reflection of hepatic ADH activity in the rates (mouse liver, 20–50 nmol/min/mg soluble protein [33]; human biopsy liver, 53–104 nmol/min/mg soluble protein [61]. Analyzing the oxygen consumption rate (O_2_ ℓ/kg/h) as a measure of the basal metabolic rate across animal species shows that it strongly correlates with the rate of alcohol metabolism; both rates are higher in animals with lower body weights (they are 6.7 times higher in mice than in humans) [37]. This is because smaller animals have a larger body surface area per kg of body weight, which leads to greater heat dissipation; therefore, smaller animals have a higher oxygen consumption rate (basal metabolic rate) to maintain body temperature, resulting in a higher alcohol metabolic rate.

In *ADH3*-knockout (*Adh3*^−/−^) mice under CAC, the Pz-sensitive (i.e., ADH1-dependent) alcohol metabolism rate increases, even though there is no increase in hepatic ADH1 protein [24,62]. This is thought to be the result of an increase in the reoxidation rate of NADH caused by CAC [35] which, in turn, increases alcohol metabolism [63,64]. A hypermetabolic state under CAC has been proposed as the cause of this increase in the reoxidation rate of NADH, in which the ADP/ATP ratio increases due to the activation of ATPase in the liver, resulting in increased mitochondrial respiratory activity [59,65]. This state of the activated respiratory chain can also be caused by exposure to cold or the administration of thyroid hormone, both of which increase oxygen consumption and alcohol metabolism due to the increase in ADP/ATP [37,60]. Drinking alcohol at high altitudes or on airplanes can also cause unexpected intoxication or hangovers due to the suppression of alcohol metabolism in an environment with low oxygen concentration; conversely, oxygen masks are effective in treating patients with acute alcohol poisoning because oxygen supply promotes alcohol metabolism [66]. Thus, the rate of oxygen consumption in mitochondria, which controls the rate of NADH reoxidation, is another important rate-limiting factor in alcohol metabolism, being linked to the hepatic ADH activity level [37].

The β value of alcoholic patients can increase to more than three times that in the general population, even though the activity of alcohol oxidases such as ADH1 or the MEOS decreases due to liver damage [67]. One of the reasons for this is thought to be the increased oxygen consumption rate caused by the hypermetabolic state under CAC. The continuation of this hypermetabolic state in alcoholic patients promotes organ damage and aging by exposing the body to oxidative stress. However, the mitochondrial respiratory chain activity that is enhanced under CAC eventually decreases as liver damage progresses [68], resulting in a decrease in the rate of alcohol metabolism.

## 3. The Roles of ADH Isoenzymes in Alcohol Metabolism In Vivo

### 3.1. Class I ADH1

ADH1 is the most representative mammalian ADH isozyme, known since the 1950s [4]. It has a low Km for EtOH of approximately 1 mm and high sensitivity against Pz inhibition (Table 1). As mentioned above, the enzymatic nature of ADH1 is reflected in the conventional ADH assay, showing maximum activity at low concentrations of EtOH (approximately 15 mM) as a substrate. ADH1 is also a key enzyme in alcohol metabolism, responsible for approximately 80% of in vivo alcohol metabolism [10,21]. However, its contribution to alcohol metabolism decreases to approximately 60% in mice with increasing EtOH concentration [21].

Although mouse ADH1 is encoded by a single gene, *Adh1*, human ADH1 is further classified into ADH1A, 1B, and 1C. These three isozymes consist of α, β, or γ subunits encoded by the genes *ADH1A, 1B,* and *1C*, respectively [27]. Furthermore, the β subunit of ADH1B contains three genetic polymorphisms (β1, β2, and β3), and the γ subunit of ADH1C contains two genetic polymorphisms (γ1 and γ2) (Table 1). ADH becomes active when these subunits are dimerized. Among them, the *ADH1B***2/*2* (β2/β2) type shows activity that is several dozen times higher than that of the **1/*1* (β1/β1) type [69,70]. However, no results suggesting that the *2*/2** type can metabolize EtOH faster than the **1/*1* type in the body were obtained in actual drinking tests at an EtOH dose of 0.4 g/kg [71]. Clinical studies have reported that, when measuring the BAC of outpatients who had drunk a large amount of alcohol the previous day, the *1*/1** type showed a significantly higher residual BAC than the *1*/2** and *2*/2** types, and the frequency of the *1*/1** type was statistically higher in alcoholics [72,73]. These reports indicate that the *ADH1B*1/*1* type metabolizes alcohol more slowly than other types and tends to retain alcohol in the body. Thus, it is suggested that the difference in enzyme activity due to the *ADH1B* gene polymorphism becomes apparent in the BAC dynamics after an individual has drunk large amounts of alcohol that remain unmetabolized until the next day, although this is not easily reflected in the BAC in drinking experiments conducted using small amounts of alcohol.

The quantity of ADH1 protein in the liver was significantly reduced in mice administered large amounts of EtOH (3–5 g/kg), resulting in a BAC of 60 mM or higher [33]. In vitro assays of ADH activity showed that mouse ADH1 activity began to decrease at EtOH levels above 15 mM due to substrate inhibition, decreasing to approximately 60% at 100 mm [20,74]. These in vivo and in vitro data on ADH1 suggest that the contribution of ADH1 to alcohol metabolism in the body decreases under high BAC.

In a CAC experiment in mice, however, ADH activity in liver extract and the quantity of ADH1 protein increased via the LD method using 4% EtOH for 1–2 months [22,46] and using 10% EtOH solution instead of water for 1 month [24,62], accompanying an increase in the rate of alcohol metabolism and mild fat accumulation in the liver. In rats administered alcohol via gavage at a dose of 13 g/kg/day for 42 days, an increase in liver ADH1 was observed at all levels of mRNA, enzyme amount, and activity [75]. However, ADH1 activity in the liver decreases as alcohol-induced liver damage progresses [50]. Thus, in response to the fluctuations in liver ADH activity, ADH1 contributes to the acceleration in the rate of alcohol metabolism by increasing its activity in the early stage of CAC, but reduces its role in alcohol metabolism under severe alcoholic liver damage by decreasing its activity.

### 3.2. Class II ADH2

ADH2 was first discovered among ADH isozymes other than ADH1 in humans, and was therefore classified as Class II [16]. Human ADH2 consists of π-ADH subunits encoded by the gene *ADH2*. As the three types of human Class I isozymes (1A, 1B, 1C) were previously called ADH1, 2, and 3, encoded by *ADH1*, *2*, and *3*, respectively [27]; ADH2 is alternatively called ADH4, encoded by the gene *ADH4* [28] (Table 1). ADH2 is exclusively localized in the liver [6] and was once considered a candidate for the non-ADH pathway because it has a higher Km and is less inhibited by Pz compared with ADH1. However, its activity is very unstable, and it shows low or no activity in approximately 30% of human livers [76] and very weak activity in rodent livers [20,77,78]. Therefore, ADH2 has not been extensively studied with regard to alcohol metabolism. Furthermore, a CAC experiment using baboons fed LD for 4 weeks reported a loss of hepatic ADH2 activity [79]. Based on these findings, the role of ADH2 in alcohol metabolism as a non-ADH1 pathway is unclear, especially in the enhancement of alcohol metabolism under CAC. However, analyses of DNA polymorphisms in the human *ADH2* promoter region have suggested that the *ADH2* genotype affects alcohol metabolism in vivo [80]; therefore, further investigations into specific conditions under which ADH2 contributes to alcohol metabolism in vivo may be necessary.

### 3.3. Class III ADH3

We identified a novel ADH in the mouse liver in 1979 [17], which differed from the well known ADH1. It has a very high Km for EtOH, an acidic isoelectric point, and high resistance against Pz. This mouse ADH corresponds to ADHB2, which was visualized as an ADH activity band on the electrophoretic zymogram of mouse ADH [7,81]. The same ADH isozyme was later identified in the human liver as χ-ADH in 1981 [18] and classified as Class III ADH because it was discovered after Class II ADH2. The gene encoding Class III ADH3 is designated *Adh3* in mice and *ADH3* in humans [27]. The human Class III ADH isozyme is also referred to as ADH5 for the same reason as Class II ADH, and is encoded by the gene *ADH5* [28].

ADH3 is the least inhibited by Pz and has the highest Km among the mammal ADH isozymes [7,18]. Furthermore, it is a multifunctional enzyme that is homologous to glutathione-dependent formaldehyde dehydrogenase (GSH/FADH) [82] and S-nitrosoglutathione reductase (GSNOR) [83]. The Harvard ADH research group, which characterized χ-ADH in humans, concluded that ADH3 did not participate in alcohol metabolism because the Km for EtOH was extremely high (over 1 M) [18]; thus, ADH3 has not been focused on in alcohol metabolism research. However, we reported that mouse ADH3 activity against EtOH was significantly increased by the addition of amphiphilic substances, which increased the hydrophobicity of the solution in the reaction medium. The solution’s hydrophobicity markedly increased the catalytic efficiency (Kcat/Km) of ADH3 for EtOH by significantly lowering its Km for EtOH [21] (Table 2). Moulis et al. also reported that human χ-ADH was activated by hydrophobic anions [84]. Additionally, we demonstrated that the solution hydrophobicity in the cytoplasm of liver cells was much higher than in the medium by using Nile red, a probe of solution hydrophobicity [21]. These findings suggest that the EtOH oxidation activity of ADH3 may be higher in vivo, which has a higher solution hydrophobicity, than in vitro.

In 1980, Barnett and Felder [85] discovered deer mice (DM) that were genetically deficient in hepatic ADH. However, although the liver ADH activity of ADH^−^DM was close to zero based on the conventional ADH assay, ADH^−^DM had more than 50% alcohol metabolism capacity compared with normal ADH^+^DM, with the rate of alcohol metabolism increasing with increasing BAC [86]. Thus, the existence of the non-ADH pathway was demonstrated in this animal species. Since then, the race to identify a candidate for the non-ADH pathway has become even more intense between the MEOS and catalase (described later). On the other hand, we proved that the liver of ADH^−^DM indeed lacked ADH1; therefore, the non-ADH pathway detected in ADH^−^DM was concretely the non-ADH1 pathway [87]. Additionally, ADH3 was shown to be present in the liver of ADH^−^DM; therefore, the liver ADH activity increased in a concentration-dependent manner with the increase in EtOH concentration due to the presence of ADH3 [87]. The liver ADH activity of ADH^−^DM at 100 mM EtOH was nearly 40% of the activity of ADH^+^DM when measured in a reaction mixture with increased solution hydrophobicity [87]. These results suggest that ADH3 may contribute to the non-ADH1 pathway in ADH^−^DM.

As mentioned earlier, liver ADH activity—measured via the conventional ADH assay using several tens of mM of EtOH as a substrate—mainly reflects the activity of ADH1; however, liver ADH activity deviates from ADH1 activity when the concentration of EtOH is increased because, in addition to ADH1, the liver extract contains ADH3, which has a high Km [74]. Although the activity of purified ADH1 in mice decreases above 15 mM of EtOH, with more than 40% inhibition at approximately 100 mM of substrate [20,74], ADH activity in liver extract exhibits only approximately 10% substrate inhibition at 100 mM [74]. Furthermore, ADH activity in the liver extract shows a stronger positive correlation with the total amount of ADH1 and ADH3 enzyme proteins than with the amount of ADH1 enzyme protein alone [33]. These results indicate that ADH activity in liver extract reflects both ADH1 and ADH3 activities. ADH2 is also present in the liver extract, but its contribution to liver extract ADH activity may be negligible due to its low activity [20].

When mice were administered 3 g/kg of EtOH, the amount of liver ADH1 protein significantly decreased, but the amount of ADH3 significantly increased [33]. These data suggest that ADH3, with a high Km, makes a higher contribution to alcohol metabolism under high BACs. In *Adh3*^−/−^ mice, a dose-dependent contribution of ADH3 to in vivo alcohol metabolism was demonstrated [21]. In humans, the kinetics of BAC show a first-order elimination curve [88,89] when BAC exceeds 100 mM during acute alcohol intoxication. This type of alcohol metabolism at high BAC may involve the action of a certain alcohol oxidase with a very high Km that cannot be explained by the MEOS or catalase, but can be explained by ADH3.

Unlike other ADH isozymes, ADH3 is distributed in all tissues and is concentrated in sinusoidal endothelial cells in the liver [90]. Therefore, ADH3 may play a barrier role in reducing the high concentration of EtOH from the digestive tract at the liver entrance and directing it to liver parenchymal cells after lowering the concentration. Baraona et al. [91] reported that sex differences in BAC in humans may be associated with sex differences in gastric χ-ADH (ADH3) activity. These findings suggest that ADH3 plays a role in the first-pass metabolism (FPM), in which ingested alcohol is metabolized in the gastrointestinal tract and liver before being transported from the heart to the systemic circulation.

ADH3 shows various adaptive changes even under CAC. When mice fed LD containing 4% EtOH for more than one month were administered a large quantity of EtOH, the quantity of ADH3 enzyme protein increased significantly in the liver [22,46], suggesting that ADH3 plays an important role in alcohol metabolism during binge drinking under chronic alcohol intake. In another CAC experiment in which *Adh1*^−/−^ mice were fed 10% EtOH solution for one month, the rate of alcohol metabolism in *Adh1*^−/−^ mice—which depends on the non-ADH1 pathway—more than doubled, reaching approximately 50% that of wild mice [24]. The quantity of hepatic ADH3 enzyme protein in the *Adh1*^−/−^ mice also increased under CAC, showing a significant positive correlation with the rate of alcohol metabolism in *Adh1*^−/−^ mice [62]. On the other hand, CYP2E1, a key component of the MEOS, did not exhibit any positive correlation with the alcohol metabolism rate in *Adh1*^−/−^ mice [24]. Thus, ADH3 is also thought to contribute to the enhancement of alcohol metabolism under CAC, together with ADH1 [24,62]. As described in Section 3.1, ADH1 activity decreases with the progression of liver damage, leading to a reduction in its role in alcohol metabolism; however, ADH3 is conversely induced by cell damage, accompanied by an increase in solution hydrophobicity [92]. In humans, liver ADH3 activity increases with the cumulative amount of alcohol consumed and remains high even in cirrhosis [50]. Therefore, ADH3 is thought to contribute to alcohol metabolism even during alcohol-induced liver damage, in conjunction with increased NADH reoxidation under a mitochondrial hypermetabolic state; thus, ADH3—which is resistant to Pz and has a high km—is thought to be responsible for the non-ADH1 pathway that enhances alcohol metabolism under high BAC and CAC, and is presumed to play an important role in alcohol metabolism under liver damage. However, the retention of ADH3’s capacity for alcohol metabolism in alcoholic patients is thought to further increase liver damage [93], which enables them to continue drinking but leads to death.

### 3.4. Class IV ADH4

As the only extrahepatic ADH, ADH4 is mainly localized in the stomach [19] and is classified as Class IV [27]. Human ADH4 consists of a σ subunit and is alternatively called ADH7; it is encoded by *ADH7* (Table 2). ADH4 is also less inhibited by Pz and has a higher Km for EtOH than ADH1 and ADH2 [20,77]. This ADH reflects the total gastric ADH activity [87] and has been postulated to contribute to the first-pass metabolism (FPM) of alcohol in the digestive tract [77,94,95]. FPM is quantified by subtracting “the area under the BAC when EtOH is administered orally (AUC_(_*_p_*_.o.)_)” from “the BAC area when the same amount of EtOH is administered intravenously (AUC_(i.v.)_).” However, this FPM is only seen when the alcohol dose is lower than 0.5 g/kg, where BAC reaches less than 10 mM [94]. In addition, the administration of cimetidine—a H^2^-blocker and a specific inhibitor of gastric ADH—induced an FPM effect in humans [95] and rats [96] who orally consumed small doses of EtOH (0.1~0.3 g/kg), in which the BAC was 3 mM or less [96]. In other experiments, however, no effects of cimetidine on FPM were observed in humans [97] and ADH^−^DM [98] who orally consumed 0.3~0.5 g/kg of EtOH. Thus, the relationship between gastric ADH and FPM is not currently clear. Other reports have suggested a contribution of ADH4 to systemic alcohol metabolism even at high BAC. For example, cimetidine administration increased AUC and decreased β values in humans and rats after approximately 20~60 mM of BAC [99], and the AUC of *Adh4*^−/−^ mice was larger than that of wild mice when they were administered 3.5 g/kg of EtOH, with BAC reaching 86 mM [100]. On the other hand, ADH4 may not contribute to the increased alcohol metabolism caused by CAC, as CAC results in a decline in gastric ADH4 activity (perhaps due to exposure to high concentrations of EtOH) and FPM [94,101]. Thus, further investigations into the role of ADH4 in alcohol metabolism in vivo, including FPM, are required.

### 3.5. Class V ADH5 and Class VI ADH6

ADH5 (alternatively called ADH6) [25] was first identified in humans and in DM [102] at the cDNA level; *ADH5* cDNA was then isolated from rats using a human probe [103]. An *ADH5* pseudogene lacking multiple exons was identified in mice [104]. Unlike other mammalian ADH isozymes, an active and stable ADH5 protein has not been isolated (Table 1).

ADH6 was identified in DM and rat livers at the cDNA level (i.e., mRNA) [26,27], but has not been found in humans or primates [105]. Therefore, human ADH isozymes are classified into five classes, while other mammalian ADH isozymes are classified into six classes, including Class VI [26] (Table 1).

## 4. The Roles of Non-ADH Pathways in Alcohol Metabolism In Vivo

### 4.1. Microsomal Ethanol-Oxidizing System (MEOS)

Alcohol metabolism that persists even under Pz administration has been recognized since the late 1960s and was named the non-ADH pathway by Lieber et al. [11]. They further proposed the microsomal ethanol-oxidizing system (MEOS) hypothesis, which states that the non-ADH pathway involves the cytochrome P450-dependent EtOH oxidation system in liver microsomes [11,15]. The MEOS is a molecular assembly of several types of P450 molecules with EtOH oxidizing ability (mainly CYP2E1), NADPH cytochrome *p*-450 reductase, and phospholipids [106,107,108]. The MEOS is resistant to Pz and has a higher Km than ADH1 (approximately 10 mM). The quantities of its proteins and EtOH oxidation activity increase with the proliferation of smooth endoplasmic reticulum (SER) in the liver due to CAC. These enzymology and intracellular structural data for the MEOS can accurately explain the dynamics of alcohol metabolism via the non-ADH pathway; therefore, the MEOS is considered the most likely candidate for the non-ADH pathway [9,10]. Shigeta et al. [109] and Takagi et al. [110,111] stated that alcohol metabolism in ADH^−^DM, which accounts for more than 50% of alcohol metabolism in normal ADH^+^DM, depends solely on the MEOS because it is not affected by the administration of the catalase inhibitor 3-aminotriazole (3AT). Furthermore, the isotope effect (D(V/K)) value of in vivo alcohol metabolism in ADH^−^DM (1.08~1.2), which was evaluated via the administration of deuterium-labeled EtOH, was closer to the D(V/K) value of in vitro EtOH oxidation by the MEOS (1.13) than by ADH (3.22) and catalase (1.83). From these data, they concluded that alcohol metabolism in ADH^−^DM is mediated by the MEOS [112]. The theory that the MEOS plays an important role in alcohol metabolism under high BAC and under CAC has been reported in over 1000 primary and secondary studies. The MEOS is now consistently described as a representative of the non-ADH pathway in textbooks and other publications on alcohol metabolism. On the other hand, Thurman et al. showed that alcohol metabolism in ADH^−^DM was inhibited within the first several hours of 3AT administration (1.5 g/kg), resulting in lowered liver catalase activity; however, both were restored after 6 h of 3AT administration. Surprisingly, MEOS activity was also strongly inhibited by the administration of 3AT, and near complete inhibition continued even after 6 h, when the inhibition of alcohol metabolism had been completely restored [113]. Furthermore, methodological problems with the isotope effect experiment were later identified [14,114], and it was suggested that an unknown ADH is involved in alcohol metabolism in ADH^−^DM [115,116]. We also reported that in addition to ADH3, a new type of ADH, which has unique Pz resistance and high Km and is absent in the ADH^+^DM liver, is present in the ADH^−^DM liver [87]. Thus, the non-ADH pathway in ADH^−^DM cannot be attributed solely to the MEOS, even if the inhibition of alcohol metabolism is not observed following the administration of the catalase inhibitor 3AT.

In experiments using *CYP2E1* KO (*CYP2E1*^−/−^) mice, no difference was observed in alcohol metabolism rates between *CYP2E1*^−/−^ and wild-type mice at various EtOH doses [117]. Furthermore, the rate of alcohol metabolism in *Adh1*^−/−^ mice via the non-ADH1 pathway showed no correlation with the amount of hepatic CYP2E, even under CAC with 10% EtOH solution for one month, although the rate more than doubled under CAC [24]. Similarly, in *CYP2E1*^−/−^ mice, the rate of alcohol metabolism under CAC using LD increased to the same level as that in wild-type mice [118]. Thus, there is no evidence that the MEOS contributes to alcohol metabolism in vivo. Even if MEOS activity in vitro is increased several fold under CAC, the activity may not be high enough to be reflected in systemic alcohol metabolism [24,118].

By the 1990s, there were no original studies on the role of the MEOS in alcohol metabolism. Dr. Lieber actively wrote reviews on the MEOS until 2005 [119,120], but did not make any significant reference to its role in alcohol metabolism after 1997 [15], instead focusing on its relationship with liver damage. Dr. Teschke, who actively researched the MEOS in collaboration with Dr. Lieber for many years, recently published a long review entitled “50 Years of MEOS Research” [121]. However, the description of the contribution of the MEOS to alcohol metabolism was not updated in the review, and there was no rebuttal to the anti-MEOS studies mentioned above.

### 4.2. Catalase–H_2_O_2_ System (Peroxisomal Catalase EtOH-Oxidizing System)

Catalase binds to H_2_O_2_ and oxidizes EtOH through its peroxidase activity. The rate-limiting factor in this reaction is the supply rate of H_2_O_2_ rather than the level of activity of catalase [12]. This enzyme is also a candidate for the non-ADH pathway, since it has a higher Km for EtOH than ADH1 and is less inhibited by Pz [13]. However, the rate of H_2_O_2_ production in the body is one order of magnitude lower than the rate of alcohol metabolism [122,123], and the administration of its inhibitor 3AT does not affect systemic alcohol metabolism [111,124]. Therefore, it has been suggested that catalase does not impact the rate of in vivo alcohol metabolism. In such a situation, it was found that the 3AT for the catalase-inhibitory effect depended on the time elapsed from 3AT administration [14], as mentioned before. When 3AT (1.5 g/kg) was administered to ADH^−^DM 1.5 h before EtOH administration (2.0 g/kg) [14,113], or when 3AT (1.0 g/kg) was administered to mice 2 h after EtOH administration (3.0 g/kg) [125], systemic alcohol metabolism was suppressed by approximately 75% and 20%, respectively. These results seem to suggest a contribution of catalase to in vivo alcohol metabolism; however, the MEOS [113] and ADH (especially ADH3) [125] were also strongly inhibited at the same levels of 3AT doses. Thus, using 3AT in investigations of the contribution of catalase to systemic alcohol metabolism was inappropriate.

H_2_O_2_, which is essential in the oxidation of EtOH by catalase, is also produced via fatty acid β-oxidation in peroxisomes; therefore, the role of Acyl CoA oxidase has been noted because it is the rate-limiting enzyme in peroxisomal H_2_O_2_ production through fatty acid β-oxidation. When rat liver was perfused with long-chain fatty acids and albumin, the rate of H_2_O_2_ production increased approximately eight-fold, and EtOH uptake in the liver also increased [126,127]. Moreover, the production of H_2_O_2_ via fatty acid β-oxidation increased under CAC [128]. In addition, when the antihyperlipidemic drug bezafibrate (0.3 g/kg) [129] was administered to rats 1 h after EtOH administration (2.0 g/kg), both peroxisomal β-oxidation and the rate of alcohol elimination from the body markedly increased [130]. In a liver perfusion experiment using 4-methylpyrazole (4-MP), an ADH inhibitor with higher specificity than Pz, a strong positive correlation was observed between EtOH uptake and the rate of H_2_O_2_ production following the addition of various fatty acids [127]. These data suggest that catalase contributes to alcohol metabolism under specific conditions. However, when ADH activity is held in the liver, catalase-dependent alcohol metabolism is strongly suppressed due to the production of NADH through ADH-dependent alcohol metabolism because H_2_O_2_ production via fatty acid β-oxidation in the peroxisome is also accompanied by NADH production [131]. Thurman et al., who proposed the hypothesis of the catalase–H_2_O_2_ EtOH oxidizing system, acknowledged that the contribution of catalase to in vivo alcohol metabolism is negative in the presence of ADH [131]. In addition, no significant decrease in the rate of alcohol metabolism was observed in C^b^s/C^b^s mice with genetically low catalase activity when EtOH was administered at various doses [117]. Thus, no solid evidence has been obtained in vivo regarding the contribution of catalase to alcohol metabolism. However, the contribution of catalase to alcohol metabolism cannot be denied under special conditions, such as ADH deficiency and the strong stimulation of peroxisome β-oxidation.

## 5. The Role of Aldehyde Dehydrogenase 2 (ALDH2) in Alcohol Metabolism In Vivo

Alcohol metabolism is inhibited when blood acetaldehyde (AcH) levels increase, because EtOH oxidation through ADH is reversible and ADH activity is inhibited by AcH. AcH in the human body is oxidized by two main aldehyde dehydrogenase isozymes, ALDH1 and 2. The genes encoding these human ALDH isozymes are located on different chromosomes, *ALDH1* on 9q21.13 and *ALDH2* on 12q24.2 [132], whereas the genes encoding human ADH isozymes (*ADH1–5*) are located on a single chromosome (chromosome 4) in a cluster (q21–25) [28]. ALDH2 in liver mitochondria controls blood AcH levels, affecting the rate of alcohol metabolism, because it has a lower km for AcH than ALDH1 in the cytoplasm [133].

Genetic polymorphisms in ALDH2 have been observed in East Asia, including the homo-active *ALDH2*1/*1*, the hetero-low active *ALDH2*1/*2*, and the homo-inactive ALDH2*2/*2. Mizoi et al. [71] reported that when Japanese adult males ingested 0.4 g/kg of EtOH, their mean blood AcH concentrations were 4.1, 23.4, and 79.3 μm, and the mean β values were 0.15, 0.13, and 0.10 mg/mL/h for the **1/*1*, **1/*2*, and **2/*2* types, respectively. These data indicate that blood AcH concentrations increase approximately 19 times in individuals with the *ALDH2*2/*2* type compared with those with the *ALDH2*1/*1* type; consequently, ALDH2 inactivity inhibits the rate of alcohol metabolism in vivo by approximately 30%.

Animal experiments have also shown that alcohol metabolism is suppressed when blood AcH concentrations rise to 20 μm or higher [53]. In addition, the in vivo alcohol metabolism rate in rats decreased when they were fed a diet that increased blood AcH concentrations by decreasing hepatic low-Km ALDH activity without changing hepatic ADH activity [35]. When Mice were acutely administered EtOH (1~4.5 g/kg), the activity of liver mitochondrial low-Km ALDH decreased; however, after several months of feeding LD containing 4% EtOH, acute administration of EtOH did not decrease or increased its activity [134]. Thus, low-Km ALDH2 activity decreased after a single drink of alcohol, which may result in increased blood AcH due to the suppression of alcohol metabolism; however, CAC adaptively increases ALDH2 activity, consequently lowering blood AcH and promoting alcohol metabolism. This result is thought to be one of the reasons why people with the *ALDH2*1/*2* genotype who initially show facial flushing after drinking alcohol become non-flushing through addictive drinking. When liver damage progresses and mitochondrial damage becomes more severe due to chronic drinking, ALDH2 activity decreases, leading to the suppression of alcohol metabolism [135].

## 6. Limitations of Research Methods for Determining Enzymatic Mechanisms Underlying Alcohol Metabolism and Future Prospects

Research on the enzymatic mechanisms underlying alcohol metabolism has been carried out by examining the relationships between the biochemical and enzymatic properties of each alcohol oxidase (e.g., Kms for EtOH, inhibitor specificities, activity changes under various conditions) and the rates of alcohol metabolism measured at various levels (isolated hepatocytes, hepatic perfusion, and individually in vivo). However, a method for accurately evaluating the contribution of enzymes to in vivo alcohol metabolism using in vitro data has not yet been developed. For example, the contribution of each alcohol oxidase to in vivo alcohol metabolism has been studied by administering specific inhibitors against each enzyme; however, the interpretation of data obtained in these studies was confusing due to the unexpectedly low specificity of inhibition. Although Pz was previously considered an ADH1-specific inhibitor, it also inhibits MEOS activity and shows different degrees of inhibition against other ADH isozymes. Additionally, 3AT—which was considered a catalase-specific inhibitor—was found to strongly suppress the activity of ADH3 and the MEOS. For these reasons, inhibitor administration methods are not appropriate for investigating the roles of enzymes in alcohol metabolism.

Experiments using KO animals are essential in evaluating the contributions of candidate enzymes to systemic metabolism. The contributions of ADH1 and ADH3 to in vivo alcohol metabolism were demonstrated using corresponding KO mice [21]. The roles of these ADHs should be confirmed using double-KO mice, re-expression of each ADH in each KO mouse, or overexpression of each ADH in wild-type mice. RNA knockdown may also be used for analyzing target enzymes, as methods using genetically engineered mice are not always successful due to compensatory adaptive changes that occur during the development process.

Where there is a large gap between in vitro enzyme activity and in vivo alcohol metabolism, it may be necessary to consider the qualitative differences between in vitro and in vivo enzyme reaction solutions. The thermal motion of water molecules in intracellular solutions is strongly constrained compared with in vitro solutions; therefore, the activity of the enzyme in vivo may differ significantly from its activity in vitro, as the solution structure greatly affects the enzyme reaction rate [136]. As mentioned earlier, ADH3 was previously considered not to contribute to alcohol metabolism in vivo because its Km for EtOH is very high (<1.0 M) compared with BAC (usually < 100 mM). However, studies using *Adh3*^−/−^ mice suggest a contribution by ADH3 to in vivo alcohol metabolism by showing a dose-dependent decrease in the alcohol metabolism rate [21]. This gap between in vitro and in vivo studies may be filled using evidence showing that ADH3 has an allosteric activation ability due to solution hydrophobicity in cells [21]. Additionally, in humans, the *ADH1B* genetic polymorphism exhibits a large difference in enzyme activities between *ADH1B**2/*2 and *ADH1B**1/*1 [69,70], but the difference in vitro is not reflected in the rate of alcohol metabolism in vivo [71]. Furthermore, no difference in the rate of alcohol metabolism is observed between *CYP2E1*^−/−^ and wild-type mice [118], despite the marked increase in MEOS activity, mainly due to CYP2E1, under CAC. These gaps between enzyme activities in vitro and the rate of alcohol metabolism in vivo may be correlated with the qualitative differences between in vitro and in vivo reaction solution media.

Differences between in vivo and in vitro data may also occur due to interactions between different enzyme molecules or different metabolic systems in vivo. For example, ADH1 activity decreases when it is s-nitrosylated by nitric oxide (NO) [137], whereas ADH3 regulates the functions of various proteins and enzymes via s-denitrosylation as s-nitrosoglutathione reductase (GSNOR) [138,139]. Therefore, it is possible that ADH3 regulates the activity of ADH1. In the livers of congenic (pure-line) *Adh3*^−/−^ mice, ADH1 activity is significantly reduced in the absence of ADH3, although the quantity of ADH1 protein does not differ from that in wild-type mice [62]. Furthermore, the level of ADH3 enzyme is lower in the livers of *Adh1*^−/−^ mice compared with wild-type mice [24]. Thus, there may be mutual regulation of the activity and quantity of protein between two ADH isozymes. The interaction between ADH-dependent and catalase-dependent alcohol metabolism mechanisms should also be considered in different metabolic systems. Catalase-dependent alcohol metabolism depends on the rate of H_2_O_2_ production in peroxisomes, where H_2_O_2_ is produced via the β-oxidation of fatty acids, accompanied by NADH production; therefore, catalase-dependent alcohol metabolism is strongly suppressed by ADH-dependent alcohol metabolism, which produces large quantities of NADH [131]. Thus, to fill the gap between in vitro and in vivo data in alcohol metabolism research, it is necessary to confirm the correlation between in vivo alcohol metabolism and candidate enzymes at different levels of activity, protein, and mRNA levels under various conditions. These interactions between enzymes and in vivo metabolism can be investigated using functional genomics—including transcriptomics, proteomics, and metabolomics [140]—which is expected to overcome the limitations of conventional approaches, enabling the exploration of gene–gene and protein–protein interactions.

The ADH/NADH reoxidation system was likely underestimated or overlooked in previous studies on alcohol metabolism, since the debate between the MEOS theory and the catalase theory focused on the non-ADH pathway; however, clarifying the contributions of new candidate enzymes will require research based on comparisons with the fundamental pathway of alcohol metabolism, the ADH/NADH reoxidation system. To prevent the MEOS and catalase from becoming a mythical being in the history of alcohol metabolism, it is important to describe the specific conditions under which they function in alcohol metabolism in vivo.

There is no doubt that a better understanding of the enzymatic mechanisms underlying alcohol metabolism in vivo will provide a foundation for further research on alcohol-related problems.

## 7. Conclusions

The liver ADH/NADH reoxidation system is the fundamental enzymatic system for alcohol metabolism in the body, which involves both ADH1 and ADH3 and may promote alcohol metabolism under CAC by activating each component. In addition, Class III ADH3 may act as a non-ADH1 pathway in vivo, especially at high blood alcohol concentrations and in alcoholic liver disease where, although the role of ADH1 in alcohol metabolism may reduce due to a decrease in its activity, the role of ADH3 may increase due to an increase in or maintenance of its activity, even in liver cirrhosis. Thus, ADH3 may compensate for the reduced role of ADH1 in alcohol metabolism. Unfortunately, ADH3 may make it possible for alcoholic patients to continue drinking heavily, leading to death. The contributions of Class II and Class IV ADHs to in vivo alcohol metabolism have also been suggested under some specific conditions but have yielded negative results regarding the enhancement of alcohol metabolism under CAC.

In the current alcohol metabolism pathway diagram, the MEOS and catalase are given the same weight as ADH, despite the lack of in vivo evidence for them (Figure 1); therefore, we proposed a new alcohol metabolism pathway diagram that presents the ADH1/ADH3/NAD reoxidation system as the main actor in alcohol metabolism, with Class I ADH1 as the lead and Class III ADH3 as a complementary background player, while the roles of the MEOS and catalase remain unclear (Figure 2).

## Figures and Tables

**Figure 1 ijms-26-09479-f001:**
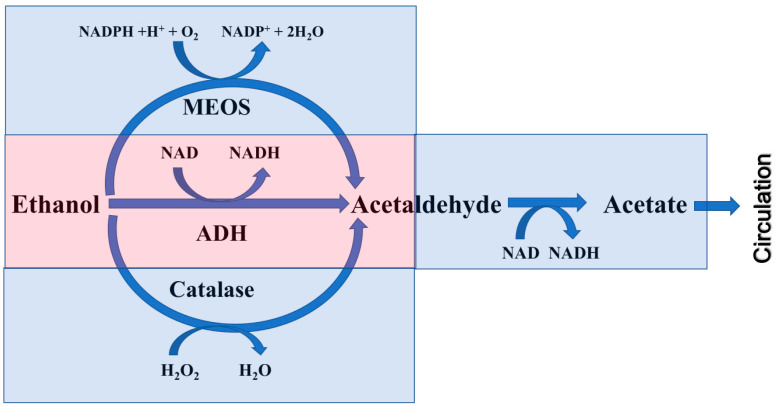
A current map of alcohol metabolic pathways in the body.

**Figure 2 ijms-26-09479-f002:**
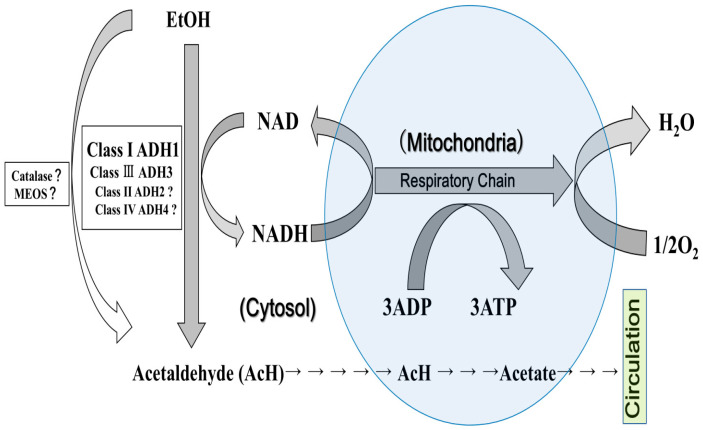
A proposal of a new map of alcohol metabolic pathways in the body.

**Table 2 ijms-26-09479-t002:** Activation of Class III ADH3-ethanolo oxidation activity by solution hydrophobicity [21].

Hydrophobic Substances	Conc. (M)	*K*_m_ (M)	*K*_cat_ (min^−1^)	*K*_cat_/*K*_m_ (M^−1^ min^−1^)
Non		NS ^a^		6.2 ^b^
tert-Butanol (C4)	0.12	3.88	37.9	9.8
	0.46	2.24	42.1	18.8
	1.38	0.31	19.4	62.6
Butyramide (C4)	0.10	3.17	26.1	8.2
	1.00	1.66	22.3	13.4
Valeramide (C5)	0.05	2.92	25.1	8.6
	0.50	1.61	24.4	15.2
Capronamide (C6)	0.10	2.38	44.6	18.7
	0.20	0.27	8.6	31.9

Activity of purified ADH3 was measured with ethanol as a substrate in the presence of a hydrophobic substance in 0.1 M Na, K phosphate buffer (pH 7.4) at 37 °C. ^a^ NS: Not saturable up to 3 M. ^b^ Estimate based on the slope of the curve of velocity vs. substrate concentration {*v* = (*K*_cat_/*K*_m_)[E][S]}.

## Data Availability

No new data were created or analyzed in this study. Data sharing is not applicable to this article.

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
