# Peer review of "Enzymatic Control of Alcohol Metabolism in the Body—The Roles of Class I, II, III, and IV Alcohol Dehydrogenases/NADH Reoxidation System, Microsomal Ethanol Oxidizing System, Catalase/H_2_O_2_ System, and Aldehyde Dehydrogenase 2"

_ijms, 2025, doi:10.3390/ijms26199479_

Round 1
Reviewer 1 Report
Comments and Suggestions for Authors
--The review is rather too long; it would benefit from restructuring and shortening by at least 1/3.
--It would be better for all abbreviations used to be listed near the start of the MS, rather than near the end.
--Please provide genetic and molecular details about the several forms of alcohol dehydrogenase and acetaldehyde dehydrogenase described: What are the chromosomal locations of the genes? What are the organizations of introns and exons? What are the sizes of the mRNAs and the protein products? Is there evidence for multiple mRNAs for some of the forms?
--A summary plot of the tissue and organ distributions of the several ADHs described would be a welcome addition.
--Although fairly well-written overall, the MS would benefit from thorough and careful redaction by an experienced medical/scientific writer whose first language is English. Some examples of things that should be changed to improve clarity and readability include the following [there are many others, too]:
--l.78--Delete 'between the two candidate alternate pathways, which was intense.'
--l.81, 84-and elsewhere--change 'non-ADH' to 'non-ADH1'
--l.99--For 'found', read 'known.'
--l. 103--change to Kms = plural--as there are many Kms.
--l.119 and elsewhere--Lieber-DeCarli.
--l.139--For 'shuttle', read 'shuttles'--as there are more than one.
--ll.205 and 208-- For 'reduce', read 'decrease.'
--l.317--For 'damages', read 'disease'.
--l.340--Re write to 'Recommended that personalized target doses of alcohol should be established for'. Is it not true actually, that the safest dose of alcohol is zero because of the growing evidence that alcohol is a risk to health? What about the usual recommendation that the dose limit for adult men should be 2 drinks per day; that for women should be 1 drink per day?
--l.401--Delete 'was found'
--l.474--For 'with hm', read 'to him to do so.'
--l.505 and elsewhere-- For 'BBV', read 'BBB'
--l.544--For 'negative', read 'negligible.'
Comments on the Quality of English LanguageAlthough fairly well-written overall, the MS would benefit from thorough and careful redaction by an experienced medical/scientific writer whose first language is English.
Author Response
Author‘s responses for Reviewer 1
Comments and Suggestions for Authors
--The review is rather too long; it would benefit from restructuring and shortening by at least 1/3.
Author’s response: According to the reviewer‘s recommendations, the MS was revised with shortening in mind, but new descriptions were added. It finally underwent the English editions by MDPI Author Service (English editing ID: english-99924).
--It would be better for all abbreviations used to be listed near the start of the MS, rather than near the end.
Author’s response: The list of abbreviations was brought to near the start of the MS, as recommended.
--Please provide genetic and molecular details about the several forms of alcohol dehydrogenase and acetaldehyde dehydrogenase described: What are the chromosomal locations of the genes? What are the organizations of introns and exons? What are the sizes of the mRNAs and the protein products? Is there evidence for multiple mRNAs for some of the forms?
Author’s response: I briefly described genetic and molecular information of ADH and ALDH isozymes, but refrained from precise information on the gene structures, the sizes of the mRNAs of these enzymes. I considered such kind of information is kept off from the content of this review and increases the MS volume.
I don’t have any information for multiple mRNAs for some of the forms of these enzymes. I would be grateful if informed.
--A summary plot of the tissue and organ distributions of the several ADHs described would be a welcome addition.
Author’s response: Table 1 was enlarged on these points within a possible scope.
--Although well-written overall, the MS would benefit from thorough and careful redaction by an experienced medical/scientific writer whose first language is English.
Author’s response: The revised MS was subjected to the English editing by MDPI Author Service before submitting.
--Some examples of things that should be changed to improve clarity and readability include the following [there are many others, too]:
--l.78--Delete 'between the two candidate alternate pathways, which was intense.'
--l.81, 84-and elsewhere--change 'non-ADH' to 'non-ADH1'
--l.99--For 'found', read 'known.'
-l. 103--change to Kms = plural--as there are many Kms.
--l.119 and elsewhere--Lieber-DeCarli.
--l.139--For 'shuttle', read 'shuttles'--as there are more than one.
--ll.205 and 208-- For 'reduce', read 'decrease.'
--l.317--For 'damages', read 'disease'.
--l.401--Delete 'was found'
--l.474--For 'with hm', read 'to him to do so.'
--l.505 and elsewhere-- For 'BBV', read 'BBB'
--l.544--For 'negative', read 'negligible.'
Author’s response: The recommended corrections were carried out. Regarding others than above, the corrections by the English editing service are expected.
--l.340--Re write to 'Recommended that personalized target doses of alcohol should be established for'. Is it not true actually, that the safest dose of alcohol is zero because of the growing evidence that alcohol is a risk to health? What about the usual recommendation that the dose limit for adult men should be 2 drinks per day; that for women should be 1 drink per day?
Author’s response: The sentence of “Guller recommended that ―, and that FPM should ― after a meal.” was delated.
Comments on the Quality of English Language
--Although fairly well-written overall, the MS would benefit from thorough and careful redaction by an experienced medical/scientific writer whose first language is English.
Author’s response: I expect an improvement of the revised MS by the English editing service I ordered.
Reviewer 2 Report
Comments and Suggestions for Authors
Your manuscript offers a valuable reexamination of enzymatic pathways involved in alcohol metabolism, particularly highlighting ADH3’s potential in vivo role, which is often overlooked in favor of MEOS or catalase. The biochemical analysis and historical synthesis are thorough, and your long-term research contributes significantly to this domain.
However, several issues need to be addressed to strengthen the manuscript’s clarity, rigor, and impact.

the manuscript can be proof read by a native English speaker
Author Response
Author‘s responses for Reviewer 2
Comments and Suggestions for Authors
Your manuscript offers a valuable reexamination of enzymatic pathways involved in alcohol metabolism, particularly highlighting ADH3’s potential in vivo role, which is often overlooked in favor of MEOS or catalase. The biochemical analysis and historical synthesis are thorough, and your long-term research contributes significantly to this domain.
However, several issues need to be addressed to strengthen the manuscript’s clarity, rigor, and impact.
Comments on the Quality of English Language
--the manuscript can be proof read by a native English speaker
Author’s response: The revised MS underwent the English editing by MDPI Author Service (ID: english-99924) before submitting.
This comprehensive and mechanistically detailed overview of alcohol review by Dr. Haseba presents an in-depth examination of the enzymatic systems involved in alcohol metabolism. The manuscript is rich in historical perspective, biochemical detail, and original experimental insights. It reassesses the long-debated roles of MEOS and catalase in vivo and strongly advocates for a renewed focus on ADH3 as a physiologically relevant contributor, especially under chronic alcohol consumption (CAC) and liver damage. The author draws from decades of personal and published work to challenge the prevailing assumptions in the field, particularly the unsubstantiated roles attributed to MEOS and catalase, and fills a significant research gap by emphasizing ADH3’s function under conditions where ADH1 activity is impaired.
Major Concerns
1) The review lacks structure as no heading or subheadings are there. In detail, the abstract is overly dense and lacks clarity for readers unfamiliar with the field. The main findings and implications should be separated out with clearer thematic focus or structured subheadings (e.g., Background, Methods, Results, Conclusions).
Author’s response: The headings in the revised MS were corrected with consideration for the structure of the review. The main findings and implications were understandably described as much as possible. The abstract was rewritten, considering the conciseness, clarity, separation of findings and implications structures.
2) As per to the review over 70% of the references (including experimental evidence and data interpretation) are from the author’s previous work (e.g., references 7, 20–24, 29, 42, 59, 83, etc.) which is commendable and shows how the authors has in depth knowledge, however, the reviewer believe independent experimental corroboration of the author's claims (e.g., ADH3's in vivo activation via solution hydrophobicity, suppression of MEOS/catalase in KO mice) is minimal. The manuscript would benefit from citing more third-party studies (e.g., recent works using proteomics, CRISPR models, or metabolomics).
Author’s response: As described in MS, it had been believed that enzymatic mechanism of alcohol metabolism is already elucidated by ADH1, MEOS and catalase, and that ADH3 does not contribute to alcohol metabolism in vivo because of its very high Km for EtOH. Consequently, there are few papers reporting on the role of ADH3 in alcohol metabolism by others than our groups. Therefore, the role of ADH3 in alcohol metabolism was referred mainly from the author’s previous works. Moulis et al. has reported the activation of human ADH3 by hydrophobic anion, similar to our results in mouse ADH3, so, their paper was newly added in the reference section.
The works using proteomics, CRISPR models, or metabolomics are expected as future prospects of the research of alcohol metabolism, as described in the revised MS.
3) The manuscript provides strong critique of MEOS and catalase based on knockout mouse models and inhibitor studies but lacks balanced consideration of contradictory reports. While ADH3 is convincingly supported, more comparative data quantifying relative contributions of ADH3 vs MEOS/catalase in various species and contexts would strengthen the argument.
Author’s response: The contribution of MEOS/catalase and ADH3 to alcohol metabolism have been exclusively investigated in rats and mice. Papers, which studied comparatively ADH3 and MOES/catalase, cannot be found other than our paper (ref. 24).
4) In Vivo Functional Evidence for ADH3 Still needs to be reinforced
Author’s response: I agree. Further investigations are required as described in section 7 of the revised MS.
5) While ADH3-KO mice are referenced (e.g., p. 8, lines 289–291), direct functional rescue experiments (e.g., ADH3 re-expression or overexpression restoring ethanol clearance in KO mice) are not shown. Suggest clarifying if these experiments exist, and if not, addressing this gap in future directions.
Author’s response: I cannot find any papers concerning direct functional rescue experiments using KO mice for not only ADH3 but also CYP2E1 or catalase. These experiments are one of the further investigations required, as described in section 7 of the revised MS.
6) The concept that “solution hydrophobicity” activates ADH3 (Table 2) is intriguing but underexplained. Is this hydrophobicity specific to cellular stress or organelles? Can it be quantified or visualized in vivo? More mechanistic clarity is needed.
Author’s response: We demonstrated that ADH3 was markedly activated for EtOH by adding various amphiphilic substances into reaction medium, which restrict the thermal motion of water molecules to increase solution hydrophobicity. The thermal motion of water molecules in the cell is also strongly suppressed by amphiphilic and other molecules in cytoplasm(ref.143). By Nile red staining, a hydrophobic fluorescent probe, we revealed that the cytoplasm in mouse liver cell has much higher solution hydrophobicity than the medium buffer by (ref. 24) (as mentioned in section 3.3 of the revised MS).
7) Discussion on Translational/Clinical Implications Is Lacking
Author’s response: In the conclusion section, I emphasized that the ADH/ NADH reoxidation system in the liver is the fundamental enzymatic metabolism of alcohol in the body, and discussed that the roles of ADH1 and ADH3 in alcohol metabolism may be changed depending on conditions including CAC. The adaptive increase in alcohol metabolism observed in heavy drinkers or alcoholic patients may be due to an activation of this system, in which the amounts of both ADH1 and ADH3 were increased and NADH reoxidation rate was activated by CAC. The continuous increase in NADH/NAD by alcohol metabolism through these ADH isozymes in liver causes fatty liver, ketosis, hyperlactacidemia, hyperuricemia, hyperglycemia, hyperlipemia. In addition, the production of acetaldehyde by these ADH isozymes produces protein adducts and lipid peroxidation. These pathological effects of alcohol metabolism through ADH are well known in the alcohol research. Therefore, the description of these clinical implications was omitted from the review in order to shorten the revised MS.
Also as described in the conclusion section, the role of ADH1 in alcohol metabolism may be reduced with progress of alcoholic liver disease due to a decrease of its activity. On the other hand, the role of ADH3 may increase due to an increase or maintain of its activity even in liver cirrhosis, and compensate for the reduced role of ADH1 in alcohol metabolism. However, the role of ADH3 in alcohol metabolism may make alcoholic patients possible to continue heavy drinking and lead them to death.
Minor Comments
- Grammar/Style: Minor grammatical inconsistencies appear throughout, such as inconsistent use of “EtOH” vs “ethanol,” or “BAC” vs “blood alcohol concentration.” Suggest harmonizing terminology.
Author’s response: I rewrote them according to the comments.
- Abbreviation Expansion: Terms like “LD method,” “ADH-DM,” and “4-MP” should be clearly expanded at first use in both the abstract and main text.
Author’s response: I rewrote them according to the comments.
- 1 and Fig. 2 Quality: Improve figure quality and annotations. They are crucial for conveying the main conceptual shifts proposed by the author.
Author’s response: Fig. 2 was rewritten within my image.
4) The reference list is thorough but heavily skewed towards older literature. Incorporating more recent reviews and primary research would make the article more relevant to the current scientific landscape.
Author’s response: Recent and primary research papers were added in the reference list, as possible as I can.
Round 2
Reviewer 1 Report
Comments and Suggestions for Authors
I am unable to find any version of the revised MS that shows the edits and changes made in light of prior review. Please submit revised with all changes tracked, as well as clean copy
Comments on the Quality of English LanguageOK for the most part, but were all recommended changes and edits made? I cannot ascertain this within any reasonable amount of time and effort.
Author Response
|
Author‘s responses to Reviewer 1 Roun 2 |
Comments and Suggestions for Authors
I am unable to find any version of the revised MS that shows the edits and changes made in light of prior review. Please submit revised with all changes tracked, as well as clean copy
Author’s response:
The manuscript was re-revised for restructuring and shortening by 1/3.
Changes were expressed by red letters.
Comments on the Quality of English Language
OK for the most part, but were all recommended changes and edits made? I cannot ascertain this within any reasonable amount of time and effort.
Submission Date
29 April 2025
Date of this review
08 Sep 2025 15:26:34
Author’s response:
The manuscript was edited on English Language again by MDPI Author Service (ID: english-100623)
